# Directed Silica Co-Deposition by Highly Oxidized Silver: Enhanced Stability and Versatility of Silver Oxynitrate

**Carla J. Spina [1],\*, Roohee Ladhani [2], Carlie Goodall [2], Michelle Hay [2] and Rod Precht [2]**

[1]  Exciton Pharma Corp, Toronto, ON M5G 1L7, Canada
[2]  Exciton Technologies Incorporation, Edmonton, AB T5J 4P6, Canada; rthaver@excitontech.com (R.L.); cgoodall@excitontech.com (C.G.); mwoodward@excitontech.com (M.H.); rprecht@excitontech.com (R.P.)
\*  Correspondence: cspina@excitontech.com

**Abstract:** Novel silver compounds in higher oxidation states, Ag (II) and Ag (III), have emerged as desirable alternatives to existing forms of antimicrobial silver compounds. Offering enhanced efficacy without sacrificing biocompatibility. Unique physiochemical characteristics associated with higher oxidation state silver confer desirable therapeutic traits. However, these same characteristics create challenges in terms of long-term stability and chemical compatibility with conventional biomedical materials. Core-shell methodologies, utilizing silica as a mesoporous or amorphous shell, have been adopted to enhance the stability of reactive active ingredients or cores. These methodologies commonly utilize controlled condensation of silicic acids in non-aqueous media by way of hydrolyzing alkyl silicates: the Stöber process or modified processes thereof. However, these strategies are not conducive to cores of higher oxidation state silver wherein hydroxyl organic precursors and by-products are incompatible with strong oxidizing agents. Addressing these challenges, we present a strategy herein for the preparation of a self-directed silver oxynitrate-silica, $Ag_7NO_{11}$:$SiO_2$, framework. The method described utilizes pH gradients generated from the oxidation reaction of soluble silver, Ag (I), with a strong oxidizing agent/alkaline silicate media to facilitate spatial control over the protonation and subsequent condensation of silicic acid from aqueous solution. The resulting $Ag_7NO_{11}$:$SiO_2$ framework confers enhanced long term and thermal stability to silver oxynitrate without impairing aqueous degradation profiles or subsequent antimicrobial and antibiofilm activities.

**Keywords:** silica; silver oxynitrate; stability; core-shell; antibiofilm; antimicrobial

---

## 1. Introduction

Silicon dioxide is a ubiquitous material in electronics, food, agriculture, and approved for use in pharmaceutical and drug delivery [1–3]. As a solid-state powder, silicon dioxide is used in solid or semi-solid formulations as a tabletting agent, carrier for active ingredients, thickener, or moisture scavenger. In core-shell structures silicon dioxide ($SiO_2$) may provide a protective shell or barrier, regulating release profiles and conferring protection or stabilization of active core materials [4,5]. Controlled growth of spherical silica particles of uniform size by means of (1) hydrolysis of alkyl silicates and (2) subsequent condensation of silicic acid in alcoholic solutions, or the Stöber process, is commonly employed in a variety of fields [6–8]. Based on this process, a number of methods for silica dioxide core-shell have been prepared from alkyl silicates, moderating the stability and release of active ingredients. This strategy has been applied to the preparation of metal and weakly oxidizing metal oxide core silica dioxide-shell materials [4,7,9]. However, the utility of this process is limited for strong oxidizing agents due to precursor and by-product compatibly. Silver has been

known for centuries to be an effective antimicrobial agent [10–12]. Throughout these centuries the foremost state or form of this antimicrobial agent has been comprised of metallic and, or singly ionic silver: Ag (0) and Ag (I) [13–15]. As a weak oxidizing agent, with a standard electrode potential of +0.7996 V, strategies for fabrication of monodisperse Ag-SiO$_2$ core-shell from Ag (I) materials have been identified [7,9]. These strategies utilize a seeded polymerization technique, based on the Stöber method, wherein sol-gel reactions of alkyl silanes generate silver nanomaterials with amorphous silica shell coatings. Protecting the silver nanoparticle cores from oxidation without hindering antimicrobial function. Recently, higher oxidation states of silver, Ag (II) and Ag (III), have come into light as a viable alternative for these predecessor lower oxidation silvers. Higher oxidation state silver compounds demonstrate superior broad-spectrum antimicrobial efficacy against microbes in planktonic and biofilm states [16–18]. Exhibiting marked efficacy against multi-drug resistant bacterium, while remaining safe and not impairing host tissue repair or function [18,19]. Ag (II) and Ag (III) are strong oxidizing agents with standard electrode potentials of +1.980 V and +1.9 V respectively [20]. Not unlike other transition metal compounds of higher oxidation states, the same properties of higher oxidation state silver compounds that elicit unique biological activity also make them susceptible to chemical reduction and thermal degradation; limiting their utilization in a wider variety of medical applications [21–23]. Furthermore, restricting their compatibility with silica core-shell processes such as the Stöber method or modified methods thereof.

Circumventing the first step in the Stöber process, the direct use of silicic acid may also be used to generate silicon dioxide as shown in Equation (1) [24,25]. However, silicic acid is known to be unstable at concentrations above 100 ppm, resulting in uncontrolled polymerization of silica gel or silicon dioxide. Silicic acid is therefore impractical as a starting material for controlled production of silicon dioxide. Higher concentration silicon solutions may be generated through hydroxide condensation of SiO$_2$, forming stable alkaline solutions of silicate salts [26,27]. In these alkali silicate solutions, $H_2SiO_4^{2-}$ and $H_3SiO_4^{-}$ salts are believed to be the dominant ions and stable in alkali solution. Decreasing the pH of the solution may result in the formation of $H_3SiO_4^{-}$ and $H_4SiO_4$ and subsequent uncontrolled formation of amorphous silica solids, as expressed in Equation (2). Approaches for controlled polymerization of silica in solution include reducing silica concentration, maintaining high pH, addition of polymerization inhibitors, and eliminating nucleation sites [5,24,27,28]. These strategies provide insight into the mechanisms for controlled silica polymerization from alkali silicate solutions, nevertheless they are also unconducive to highly oxidized silver due to the reactivity of silver with organic reagents and reaction by-products. Accordingly, methods for the preparation of silica-encapsulated highly oxidized silver compounds to enhance stability while facilitating antimicrobial efficacy are needed.

$$SiO_2 + 2H_2O \Leftrightarrow H_4SiO_4 \tag{1}$$

$$H_3SiO_4^{-}{}_{(aq)} + H_4SiO_{4(aq)} \rightarrow (OH)_3Si\text{-}O\text{-}Si(OH)_{3(s)} + OH^{-}{}_{(aq)} \tag{2}$$

In this paper, we present a facile, one-pot method for the preparation of a higher oxidation state silver-silica gel, $Ag_7NO_{11}:SiO_2$, framework based on the direct oxidation of silver nitrate from an oxidizing alkali silicate aqueous solution. The corresponding characterization, thermal stability, aqueous degradation, and antimicrobial efficacy of the $Ag_7NO_{11}:SiO_2$ framework are evaluated over a range of relative silica concentrations based upon the one-pot reaction described herein.

## 2. Materials and Methods

### 2.1. Materials

Silver nitrate (AgNO$_3$, ACS reagent grade, ≥99.0%), nitric acid (HNO$_3$, ACS reagent grade, 68–70%), and acetone ((CH$_3$)$_2$CO, ACS reagent grade, 99.5%) were obtained from Sigma-Aldrich, St. Louis, Missouri, United States; potassium persulfate (K$_2$S$_2$O$_8$, ACS reagent grade, ≥99.0%) and sodium chloride (NaCl, ACS reagent grade, ≥99.0%) were obtained from VWR, Mississauga,

Ontario, Canada; and silicic acid potassium salt solution ($K_2SiO_3$, 39.2 wt/wt%) was obtained from PQ Corporation, Malvern, Pennsylvania, United States. All reagents were used without further purification. Unless otherwise mentioned, reverse osmosis (RO) water was used for all experimental procedures. *Staphylococcus aureus* (ATCC 6538) and *Pseudomonas aeruginosa* (ATCC 9027) strains were obtained from the American Type Culture Collection (ATCC), Manassas, Virginia, United States. Cultures were stored at ≤−70 °C and propagated on Mueller-Hinton agar (MHA) (at 37 °C for 24 h) immediately prior to experimentation.

## 2.2. Equipment

Scanning electron microscopy (SEM) was performed on a FEI Quanta FEG 250 ESEM (Thermo Scientific, Waltham, Massachusetts, United States) variable pressure and environmental scanning instrument, housed in the Centre for Nanostructured Imaging at the University of Toronto. SEM was performed in a low vacuum 70–130 Pa, imaging at 5–10 eV. Energy-dispersive X-ray spectroscopy (EDX) was performed on the same FEI Quanta FEG 250 ESEM instrument under equivalent conditions. Post-processing EDX analysis was performed using TEAM: Texture & Elemental Analytical Microscopy software. Transmission Electron Microscopy (TEM) was performed on a H-7000 TEM (Hitachi, Chiyoda, Tokyo, Japan), housed in the Microscopy Imaging Laboratories at the University of Toronto. TEM was performed under high vacuum at 75.0 kV. No coating methods were employed in performing TEM or SEM imaging. X-ray diffraction (XRD) was performed on a D2 Phaser (Bruker, Billerica, Massachusetts, United States) powder X-ray diffractometer, housed at Exciton Technologies Inc. in Edmonton Alberta. XRD was performed using Cu Kα 1.54060 A, divergence slit 0.6 mm, air scatter shield 3 mm, air scatter slit 8 mm, step size 0.010°, step time 42 sec. XRD data analysis, including peak area and full width half max (FWHM) calculations was performed using DIFFRAC EVA V4.1.1 software, post-processing including stripping Cu K$\alpha_2$.

## 2.3. Preparation of the Ag$_7$NO$_{11}$:SiO$_2$ Framework

A series of silver oxynitrate-silica co-deposition products, wherein the molar ratio of Ag:SiO$_2$ were varied from 1:0.0 to 1:0.5, were prepared as follows. Aqueous stock solutions of silver nitrate (AgNO$_3$, 59.3 wt/wt%) were prepared immediately prior to synthesis. To an aqueous solution of K$_2$S$_2$O$_8$ (7.52 mmol), 0.0 to 5.21 mmol of K$_2$SiO$_3$ was added dropwise at 25 °C. The clear, colourless aqueous solution was stirred for five minutes. To this stirring solution, 1.14:1 molar equivalents of silver (AgNO$_3$:K$_2$S$_2$O$_8$), from the stock silver nitrate solution, was added dropwise into the vortex of the stirring solution, completing the addition of silver nitrate over the course of 60 s. A 50 mL total volume for the reaction solution was conserved. Over the course of silver addition, the solution transitioned rapidly from red-brown to black turbid solution in the absence of alkali silicates and from a clear bright yellow color through a translucent orange-red to a turbid brown-black suspension over the course of three to five minutes in the presence of alkali silicates. Following complete silver nitrate addition, mechanical stirring continued for a total time of 40 min. The appearance of the solution did not change during this mixing time, where the final solution appeared black and turbid. Subsequent to the 40 min of reaction time, the black suspension was filtered through a Whatman no. 40 ashless filter paper under 22 mmHg vacuum filtration. The resulting grey-black powder was washed three times with water and was subsequently rinsed three times with acetone. The product was dried under air until a steady mass of the grey-black dull powder products was obtained. Powder Ag$_7$NO$_{11}$:SiO$_2$ products were characterized by XRD, SEM, TEM, EDX, and potentiometric titration for silver content determination.

## 2.4. Silver Quantification

Quantitative determination of the percent silver content by mass for silver oxynitrate and the Ag$_7$NO$_{11}$:SiO$_2$ was completed in triplicate by potentiometric titration against sodium chloride (NaCl). In brief, samples (0.25 g) of the higher oxidation state silver compounds were digested in approximately 25 mL of dilute nitric acid (1:4, HNO$_3$:H$_2$O) overnight to liberate all silver into free Ag (I) ions in

solution. This digested solution was then quantitatively transferred to a 250 mL volumetric flask and diluted with RO water. A 5.0 mL aliquot of this solution was quantitatively transferred to a 50 mL sample vial containing 10 mL of dilute nitric acid (1:4, $HNO_3$:$H_2O$) and 20 mL water. The potential is measured across the analyte solution over the duration of titration with a 0.1 M NaCl titrant. The silver content of the sample volume is determined from the second derivative of the potential voltage plotted versus titrant volume, at an equimolar ratio of silver to chloride. In this manner, the mass percent silver content for silver oxynitrate and the grey $Ag_7NO_{11}$:$SiO_2$ powders was determined.

## 2.5. X-Ray Diffractometry

The crystalline structures of silver oxynitrate and $Ag_7NO_{11}$:$SiO_2$ were determined by X-ray diffractometry (XRD). In brief, approximately 0.5 g of either the silver oxynitrate or $Ag_7NO_{11}$:$SiO_2$ powder was spread evenly into the depression in the powder XRD sample holder and placed into the X-ray diffractometer and measured from 10 to 90 °2$\Theta$ recorded over 30 min. From crystal lattice structure diffraction patterns, the crystalline solid-state composition of the samples was determined. The silver oxynitrate and other solid-state compounds were identified using XRD spectra deferring the Crystallography Open Database (COD) [21,23,29]. The relative compositions of the primary silver oxynitrate species and by-product or degradation products were approximated using the percent relative peak height of silver oxynitrate (36.3 °2$\Theta$, COD Card 2310073) versus the additional solid-state compounds identified: AgO (32.4 °2$\Theta$, COD Card 1509488), $Ag_2O$ (32.8 °2$\Theta$, COD Card 4318188), Ag (38.0 °2$\Theta$, COD Card 1100136), $AgNO_3$ (29.7 °2$\Theta$, COD Card 1509468), and $Ag_2SO_4$ (28.1 °2$\Theta$, COD Card 1509700). Over a four-month time course, the relative solid crystalline composition of each of the samples was observed. In this manner, the stability of the higher oxidation state silver compounds was evaluated under both ambient or room temperature and at elevated temperature, 40 °C, in a temperature regulated incubator.

## 2.6. Aqueous Decomposition Studies

The decomposition profile of silver oxynitrate and $Ag_7NO_{11}$:$SiO_2$ was evaluated in aqueous media over a course of seven days. In brief, approximately 1 g of silver oxynitrate or $Ag_7NO_{11}$:$SiO_2$ was added to a series of sealed glass vials denoted 2 h, 6 h, 24 h, 72 h, 120 h, and 168 h, each containing 10 mL water. Following addition of the higher oxidation state silver compounds, the vials were sealed and vortexed for 30 s to disperse the solids in solution then stored at room temperature away from direct light. At each specified time interval, the entire contents of the vial were quantitatively transferred to a Buchner funnel with Whatman 42 ashless filter paper. The grey-black solids were washed three times with water and was subsequently rinsed three times acetone. The solids were dried under air until a steady mass was obtained. At each specified time point, powder X-ray diffraction and scanning electron microscopy, as described above, was performed on the isolated solids.

## 2.7. Planktonic Log Reduction Assay

The planktonic antimicrobial activities of the higher oxidation state silver compounds were evaluated for their efficacy against *Staphylococcus aureus* (ATCC 6538) and *Pseudomonas aeruginosa* (ATCC 9027). Briefly, the equivalent of 10 mg silver (Ag) of the higher oxidation state silver compounds were weighed and added into a triplicate set of 15 mL sealed tubes containing 10 mL Mueller Hinton Broth (MHB) challenged with $1 \times 10^6$ CFU/mL inoculum of either *S. aureus* and *P. aeruginosa.* Testing included negative control tubes without any added silver compounds. Sterility control were also included; not containing any silver compounds and not inoculated. The test and control tubes were incubated at 37 °C for four (4) hours for *S. aureus* or one (1) hour for *P. aeruginosa.* After the final inoculation and at each required incubation time, contents of the reaction tubes were neutralized with 0.4 wt/wt% sodium thioglycolate solution (STS), serially diluted with 0.89 wt/wt% NaCl (Saline) and plated onto Mueller Hinton Agar (MHA). Plates were incubated for 18–24 hrs and then enumerated.

Log reduction of the higher oxidation state silver compounds as compared to the control reaction tubes were calculated.

### 2.8. Biofilm Log Reduction Assay

Biofilm of *Staphylococcus aureus* (ATCC 6538) and *Pseudomonas aeruginosa* (ATCC 9027) were grown in a three-dimensional matrix to determine the efficacy of higher oxidation state silver compounds against bacterial biofilm. In, brief, three to five layers of sterile cotton gauze were placed in simulated wound fluid (SWF) in 6-well tissue culture plates for each strain tested. To each well, a $1 \times 10^6$ CFU/mL inoculum was added every 24 hrs for up to 72 hrs. During which time the plates were incubated at 37 °C with shaking at 200 rpm. After the incubation period, the gauze was removed from the liquid culture medium. The gauze biofilm was rinsed three times with sterile water to eliminate planktonic bacterium and then placed onto the surface of an MHA plate (Oxoid, Nepean, ON, Canada). The gauze biofilm was overlaid with additional MHA cooled to ca. 50 °C such that one-half of the biofilm was embedded in the agar and one-half was exposed. An equivalent of 10 mg silver (Ag) of the higher oxidation state silver compounds were weighted over-laid onto the biofilm and exposed to treatment at 37 °C for four (4) hours for *S. aureus* or two (2) hour for *P. aeruginosa*. After the exposure time, the dressings and biofilm (gauze pieces) were carefully removed from the plates and were placed into 10 mL of 0.4 wt/wt% STS. They were vortexed ($3 \times 1$ min) to disrupt the biofilm, then serially diluted in saline (0.89 wt/wt% saline) and spot-plated onto MHA for viable cell counts. Negative controls were made as follows: (i) gauze biofilm were grown and were overlaid with agar as described above; and (ii) no higher oxidation state silver compounds were overlaid on the biofilm in the same manner as the treatment conditions. Log reduction of the higher oxidation state silver compounds versus the control biofilm were calculated.

### 2.9. Statistical Analysis

Data are expressed as mean ± standard deviation of at least three independent experiments. Analysis of the data distribution was performed using Student's *t*-test to analyse the significance of differences between the treated group and the control group (without silver exposure). *p* values of less than 0.05 were considered statistically significant.

## 3. Results

### 3.1. Characterization of $Ag_7NO_{11}$:$SiO_2$

Synthesis of higher oxidation states of metals, such as silver oxynitrate, may proceed through the addition of a soluble metal to a strong oxidizing agent [30]. Aqueous oxidization of silver nitrate by potassium persulfate, as per Equation (3), results in the formation or deposition of $Ag_7NO_{11}$.

$$7AgNO_{3\ (aq)} + K_2S_2O_{8\ (aq)} + 8H_2O_{(l)} \rightarrow$$
$$Ag_7O_8NO_{3\ (s)} + 6HNO_{3\ (aq)} + H_2SO_{4\ (aq)} + K_2SO_{4\ (aq)} + 4H_{2\ (g)} \tag{3}$$

Based on the methods described herein, a series of silver oxynitrate-silica co-deposition products were prepared through the addition of an alkaline potassium silicate ($K_2SiO_3$) to a potassium persulfate solution prior to silver nitrate addition. Within the series of co-deposition products, the molar equivalents of silicon dioxide to silver were incrementally augmented following the series: 0:1, 0.1:1, 0.25:1, and 0.5:1 molar equivalents $SiO_2$:Ag; these co-deposition products are herein referred to as $Ag_7NO_{11}$:$SiO_2$. Subsequent to their isolation, the relative solid-state composition of the powders was determined by X-ray diffractometry (XRD). Principal diffraction patterns collected from each the powders, shown in Figure 1, were in agreement with the crystallographic parameters for $Ag_7NO_{11}$ [29]. Minor diffraction patterns were identified in each of the powder samples in agreement with the crystallographic parameters for silver sulfate. No additional diffraction patterns were identified in any of the powder samples. Silica gels formed from alkali silicates are commonly amorphous and poorly

crystalline and therefore are expected to have limited or indiscernible diffraction patterns [31]. The relative diffraction peak area for silver sulfate $Ag_2SO_4$ (28.1 °2Θ) versus silver oxynitrate (36.3 °2Θ) was identified to increase with the relative molar ratio of $SiO_2$:Ag, as shown in Figure S1. Scanning electron microscopy (SEM) and transmission electron microscopy (TEM) were employed to observe the structural composition and distribution of silver oxynitrate and silica within the isolated products. SEM images of silver oxynitrate control and $Ag_7NO_{11}$:$SiO_2$ prepared at different silica concentrations ranging from 0:1 to 0.5:1 molar equivalents $SiO_2$:Ag are shown in Figure 2. It is observed from these images that distinct cuboctahedron structure of silver oxynitrate is conserved from 0:1 to 0.25:1 $SiO_2$:Ag, where the silica is observed as amorphous structures. At 0.5:1, $SiO_2$:Ag solid silica gel is the primary structure observed and the geometric silver oxynitrate structures are not visibly detected. Upon increasing the relative concentration of silica, it is also observed that the relative crystalline size of the silver oxynitrate decreases from 0:1 to 0.5:1 $SiO_2$:Ag. This reduction in crystalline size upon increasing silica concentration was also corroborated by XRD, employing the Debye-Scherrer equation, as shown in Figure S2 [32]. Complimentary to the SEM images, TEM provides a spatial orientation of the silver within the silica framework as the density disparity between silver oxynitrate and silica affords clear demarcation between the materials as shown in Figure 3. At a low relative concentration of silica, 0.1:1, molar equivalents $SiO_2$:Ag, the distinct cuboctahedra structure of silver oxynitrate is observed as a black silhouette, Figure 3A–C. Surrounding silver oxynitrate, the less electron dense structure of silica is observed on the structural faces of silver oxynitrate, Figure 3B, and encasing the silver oxynitrate particles in a framework of silica gel, Figure 3A. Increasing the relative concentration of silica to 0.5:1 molar equivalents $SiO_2$:Ag, as shown in Figure 3D–F, the distinct cuboctahedra structure of silver oxynitrate is partly conserved. Additionally, sub-micron sized areas of high electron density within the silica gel framework are observed, Figure 3D,E. Presence of silver in these areas of high electron density in $Ag_7NO_{11}$:$SiO_2$ containing 0.5:1 molar equivalents $SiO_2$:Ag was confirmed using energy dispersive X-ray spectroscopy as shown in Figure S3. Confirming that the silica framework contains 55 to 65 wt/wt% silver, in good agreement with quantitative silver titration results shown in Figure S4.

### 3.2. Ambient and Accelerated Thermal Stability of $Ag_7NO_{11}$:$SiO_2$

Diffraction patterns of silver oxynitrate and $Ag_7NO_{11}$:$SiO_2$ containing 0.1:1 $SiO_2$: were evaluated over a period of four months (16 weeks) under both ambient and elevated temperature storage conditions (40 °C). Within this timeframe, degradation of solid-state silver oxynitrate was evaluated by monitoring the depletion of the standard $Ag_7NO_{11}$ diffraction peaks: 31.3, 36.3, and 39.7 °2Θ and the onset of the first thermal decomposition product argentic oxide, Equation (4), following standard AgO diffraction patterns peaks: 32.2, 32.4, 37.3, and 38.6 °2Θ [21,23]. For the purposes of this study, the silver oxynitrate peak at 36.3 °2Θ peak was utilized to distinguish the presence/absence of silver oxynitrate as there are no overlapping impurity or degradation product diffraction peaks at this diffraction angle. At the study commencement, minor silver sulfate ($Ag_2SO_4$) impurities were additionally identified using standard diffraction patterns peaks: 28.1, 31.2, 33.9 °2Θ.

$$Ag_7NO_{11} \rightarrow 6AgO + AgNO_3 + O_2 \tag{4}$$

Following one month of silver oxynitrate storage under ambient conditions, Figure 4A, the primary solid-state compound observed was $Ag_7NO_{11}$ with minor peaks attributed to AgO and $Ag_2SO_4$. Following four months of storage at room temperature, AgO and $Ag_2SO_4$ alone were observed, consistent with the first thermal decomposition step shown in Equation (4). In parallel, the stability of $Ag_7NO_{11}$:$SiO_2$ containing 0.1:1 $SiO_2$:Ag was evaluated under ambient conditions as shown in Figure 4B. Silver oxynitrate was identified as the primary solid-state compound following one month of storage under ambient conditions with minor peaks attributed to AgO and $Ag_2SO_4$. Similarly, at four months of storage under ambient conditions, the primary solid-state compound observed was $Ag_7NO_{11}$ with only minor peaks attributed to AgO and $Ag_2SO_4$.

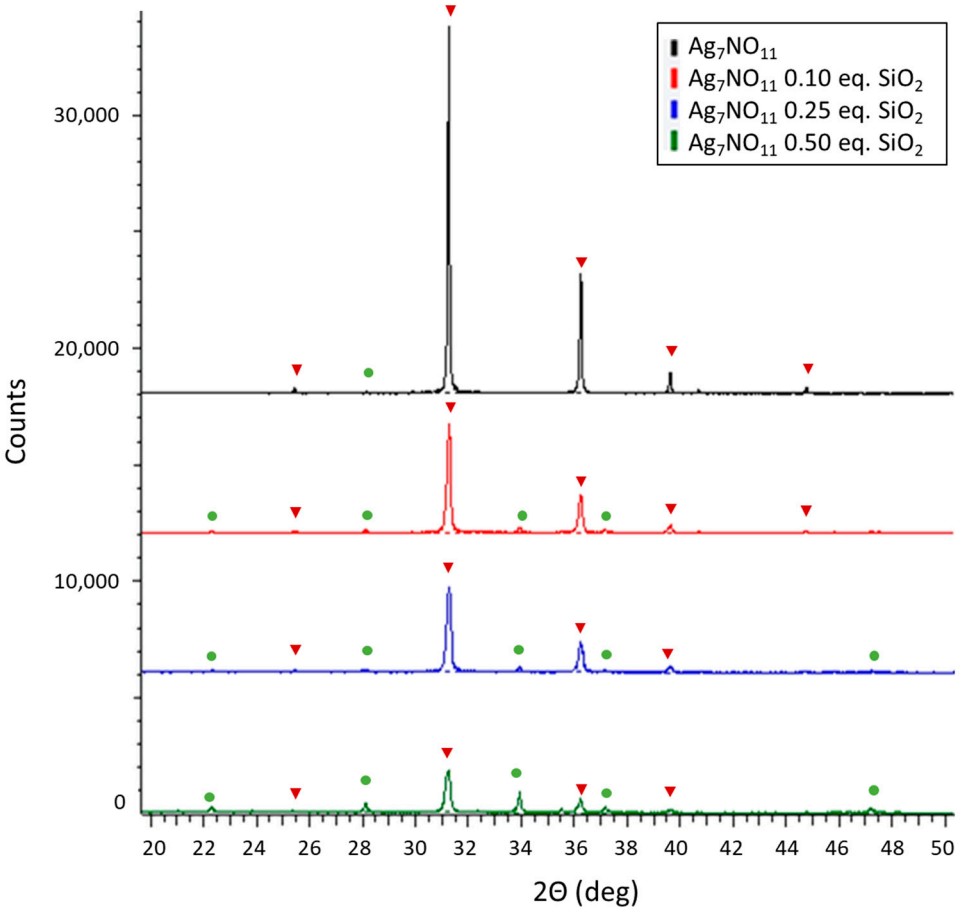

**Figure 1.** Powder X-ray diffraction of $Ag_7NO_{11}$:$SiO_2$ (0.0:1 to 0.5:1 molar equivalents $SiO_2$:Ag). Solid-state silver compounds were identified as silver oxynitrate ($Ag_7NO_{11}$, inverted red triangles) and silver sulfate ($Ag_2SO_4$, green circles).

Storage of silver oxynitrate under elevated temperatures, Figure 5A, was observed to result in multi-stage thermal decomposition processes as shown in Equations (5) to (7) [21,23]. Within one week of storage under 40 °C, the predominant solid-state compound was identified as silver nitrate, in good agreement with standard $AgNO_3$ diffraction peaks: 29.7, 32.8, and 35.3 °2Θ. Secondary solid-state compounds were identified as $Ag_7NO_{11}$, AgO, and $Ag_2SO_4$. Following 16 weeks of storage at elevated temperatures, the diffraction peak at 36.3 °2Θ for $Ag_7NO_{11}$ was not observed nor were any standard diffraction patterns for AgO. The primary solid-state diffraction patterns identified in the XRD pattern at 16 weeks were in agreement with $AgNO_3$, $Ag_2O$, and Ag (metallic silver), indicative of advanced thermal decomposition pathways as per Equations (5) to (7). In contrast, decomposition of $Ag_7NO_{11}$:$SiO_2$, containing 0.1:1 $SiO_2$:Ag, under elevated temperatures was less severe. Retaining $Ag_7NO_{11}$ as the predominant solid-state compound with only minor peaks attributed to AgO and $Ag_2SO_4$ after one week and conversion to the first thermal decomposition product, AgO, following 16 weeks of storage under 40 °C.

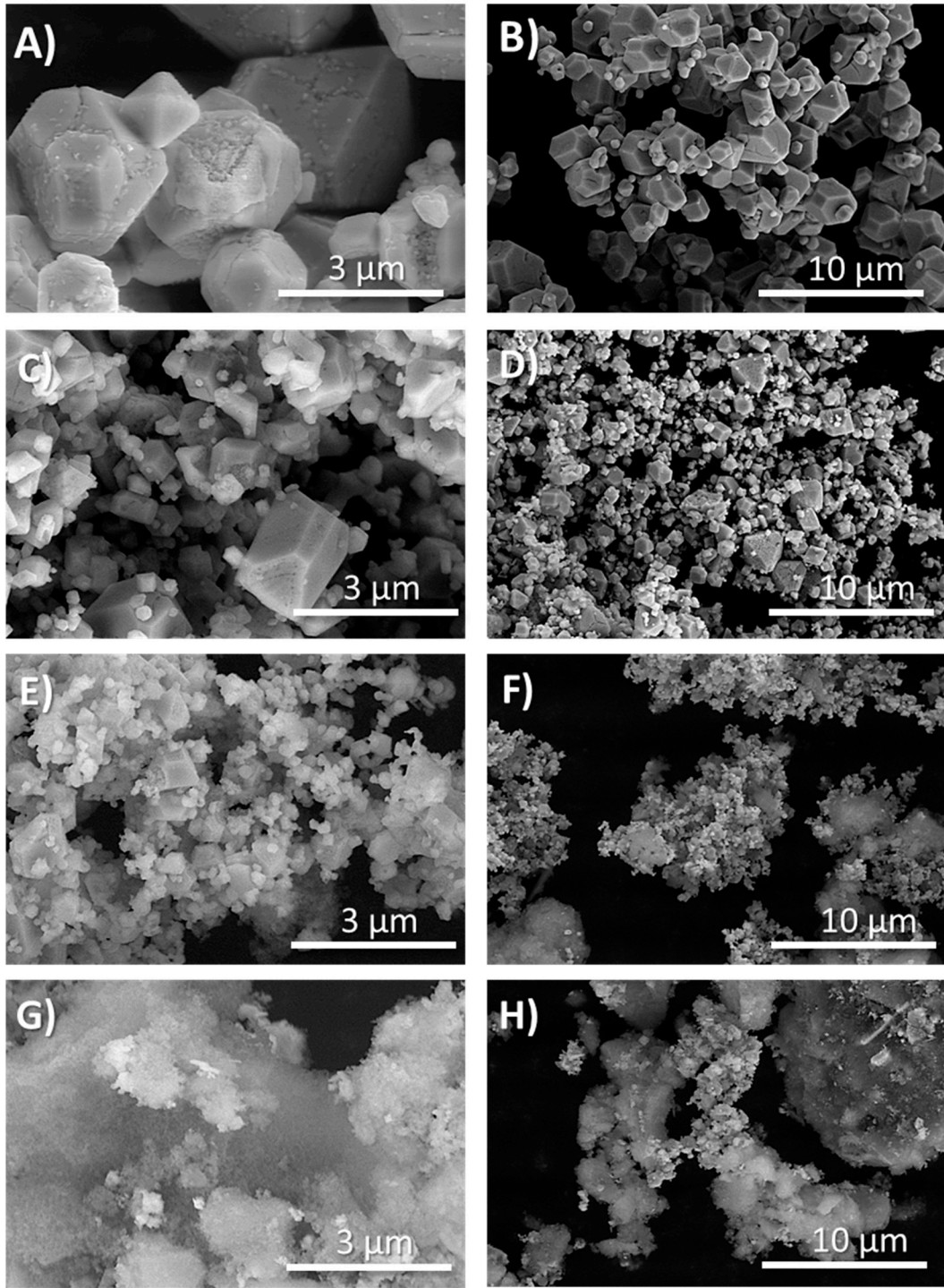

**Figure 2.** Scanning electron microscopy images of $Ag_7NO_{11}$:$SiO_2$ (0.0:1 to 0.5:1 molar equivalents $SiO_2$:Ag). (**A**,**B**) Silver oxynitrate ($Ag_7NO_{11}$); (**C**,**D**) $Ag_7NO_{11}$:$SiO_2$ containing 0.10:1, $SiO_2$:Ag; (**E**,**F**) $Ag_7NO_{11}$:$SiO_2$ containing 0.25:1, $SiO_2$:Ag; (**G**,**H**) $Ag_7NO_{11}$:$SiO_2$ containing 0.50:1, $SiO_2$:Ag.

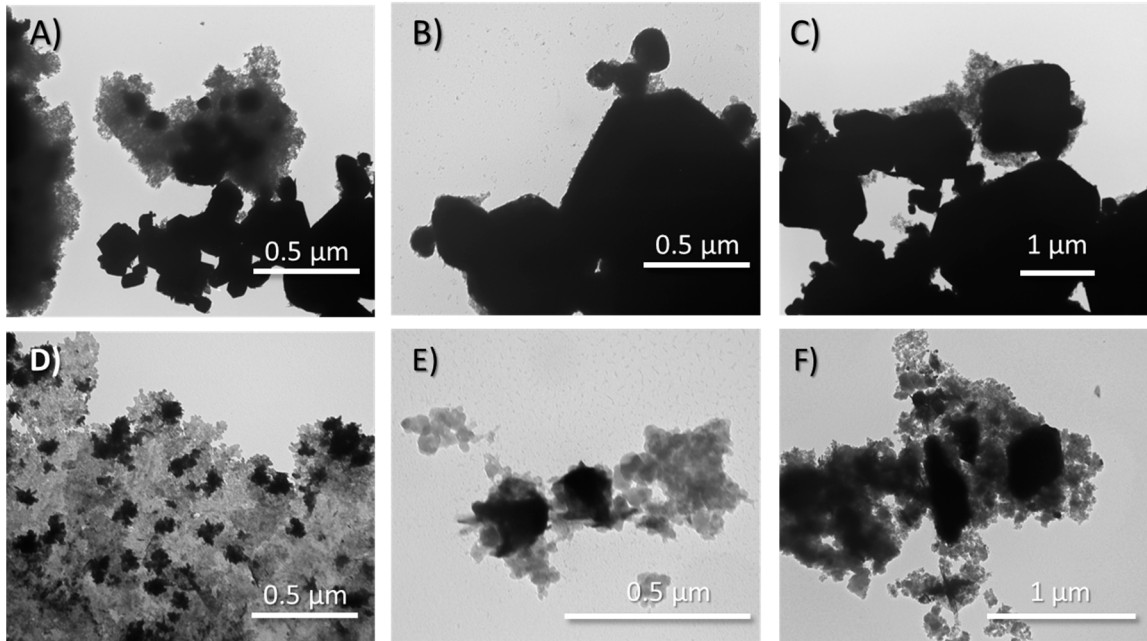

**Figure 3.** Transmission electron microscopy images of $Ag_7NO_{11}$:$SiO_2$ (0.0:1 to 0.5:1 molar equivalents $SiO_2$:Ag) prepared by the co-deposition synthetic process. (**A–C**) $Ag_7NO_{11}$:$SiO_2$ containing 0.10:1, $SiO_2$:Ag; (**D–F**) $Ag_7NO_{11}$:$SiO_2$ containing 0.50:1, $SiO_2$:Ag.

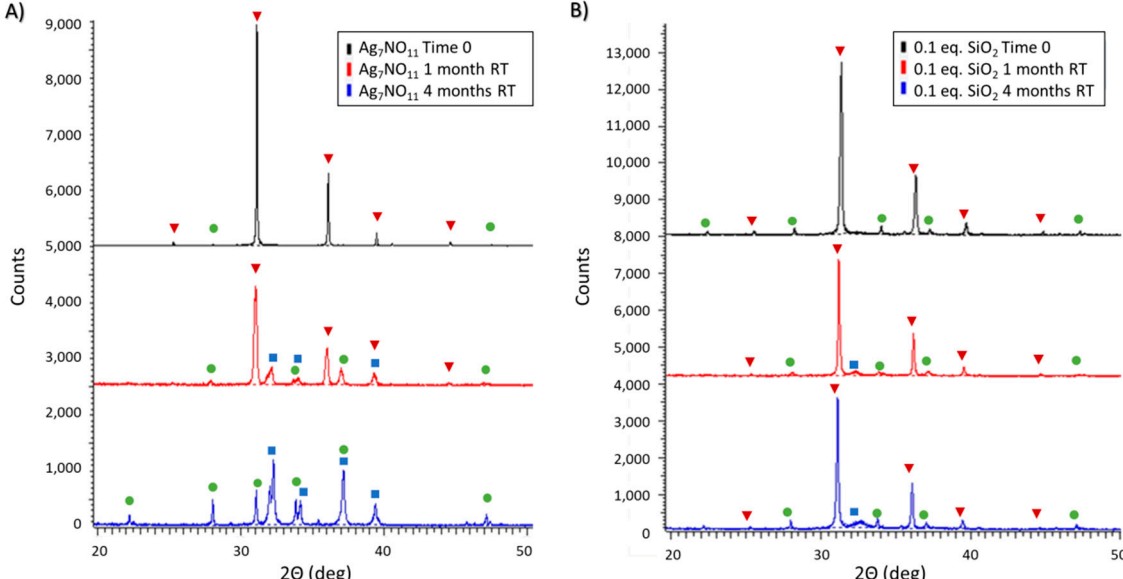

**Figure 4.** Powder X-ray diffraction patterns of (**A**) silver oxynitrate ($Ag_7NO_{11}$) and; (**B**) $Ag_7NO_{11}$:$SiO_2$ containing 0.10:1, $SiO_2$:Ag, equivalents silica over a four month period under storage at ambient room temperature (RT) conditions. Solid-state silver compounds were identified as silver oxynitrate ($Ag_7NO_{11}$, inverted red triangles), silver sulfate ($Ag_2SO_4$, green circles), and argentic oxide (AgO, blue squares).

$$6AgO + AgNO_3 \rightarrow 3Ag_2O + AgNO_3 + 1.5O_2 \tag{5}$$

$$3Ag_2O + AgNO_3 \rightarrow 3.5Ag_2O + NO_2 + 0.25O_2 \tag{6}$$

$$3.5Ag_2O \rightarrow 7Ag + 1.75O_2 \tag{7}$$

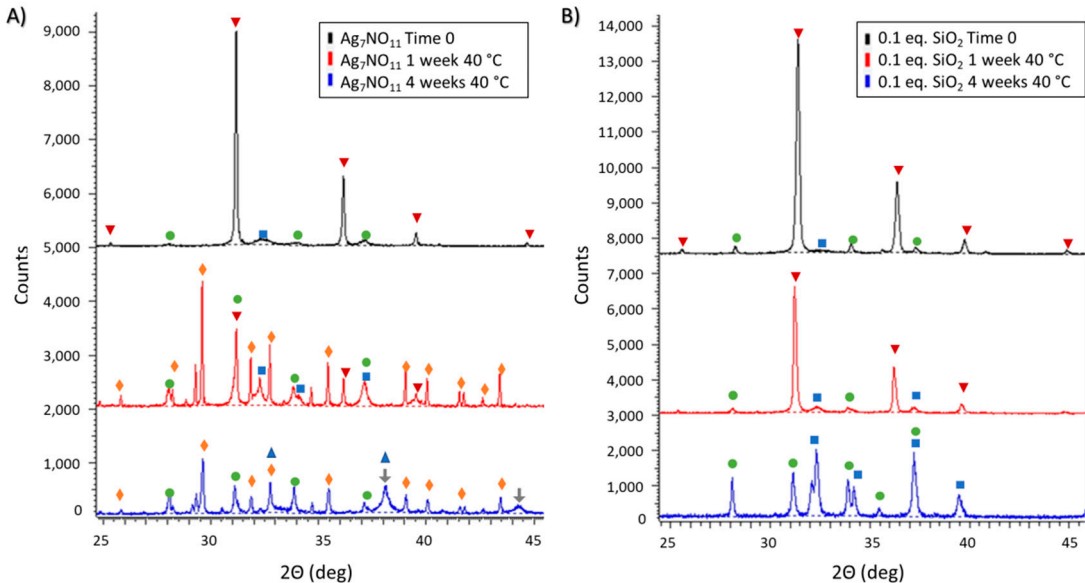

**Figure 5.** Powder X-ray diffraction patterns of (**A**) silver oxynitrate ($Ag_7NO_{11}$) and; (**B**) $Ag_7NO_{11}$:$SiO_2$ containing 0.10:1, $SiO_2$:Ag, equivalents silica over a four month period under accelerated storage in an incubator at 40 °C. Solid-state silver compounds were identified as silver oxynitrate ($Ag_7NO_{11}$, inverted red triangles), silver sulfate ($Ag_2SO_4$, green circles), argentic oxide (AgO, blue squares), silver nitrate ($AgNO3$, orange diamonds), silver oxide ($Ag_2O$, blue triangle), and metallic silver (Ag, grey arrows).

### 3.3. Aqueous Decomposition of $Ag_7NO_{11}$:$SiO_2$

The aqueous decomposition profiles of silver oxynitrate and $Ag_7NO_{11}$:$SiO_2$ containing 0.1:1 $SiO_2$:Ag were also investigated. In brief, the solid powder products were dispersed into aqueous media at room temperature and, at set time intervals over seven days, were evaluated by XRD and SEM. Following dispersion into aqueous media it was observed that, with increasing relative ratios of silica, the $Ag_7NO_{11}$:$SiO_2$ powders had an increased time of suspension as shown in Figure S5. The aqueous decomposition profiles were observed to be similar for silver oxynitrate and $Ag_7NO_{11}$:$SiO_2$ as seen in Figure 6. Within two hours of exposure to water, silver oxynitrate was confirmed as the primary solid-state silver species, with minor components attributed to AgO and $Ag_2SO_4$ confirmed by means of their respective standard diffraction patterns. Following 24 h in aqueous media, silver oxynitrate remained the primary solid-state compound, with minor components attributed to AgO and $Ag_2SO_4$. Relative counts for AgO versus $Ag_7NO_{11}$ were proportionally higher in $Ag_7NO_{11}$:$SiO_2$. At the terminus of the seven-day study, AgO was identified as the primary solid-state species for both silver oxynitrate and $Ag_7NO_{11}$:$SiO_2$, with secondary diffraction patterns attributed to $Ag_2SO_4$. The recovered solids at each time point were imaged by SEM. Typical geometric crystalline structures of silver oxynitrate were observed in Figure 7A,D for both silver oxynitrate and $Ag_7NO_{11}$:$SiO_2$. Subsequent to aqueous exposure, a crystalline morphology transformation was observed in Figure 7C,F previously identified to be associated with the formation of argentic oxide, AgO [22]. Retention of the silica framework in the co-deposition product was observed over the seven-day evaluation period as seen in Figure 7E,F.

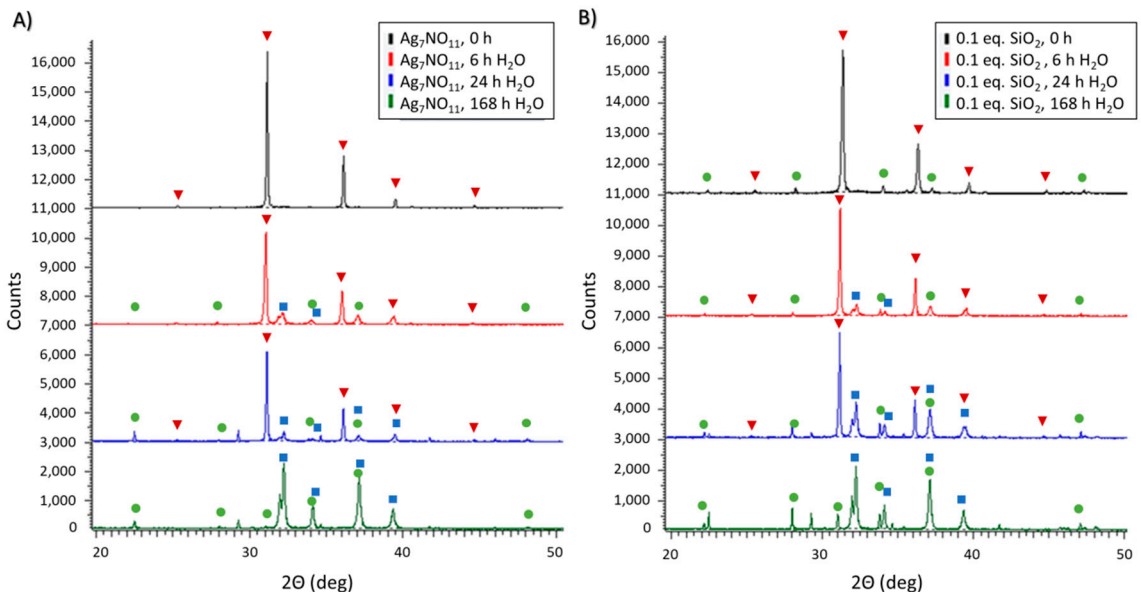

**Figure 6.** Powder X-ray diffraction patterns of (**A**) silver oxynitrate (Ag$_7$NO$_{11}$) and; (**B**) Ag$_7$NO$_{11}$:SiO$_2$ containing 0.10:1, SiO$_2$:Ag, equivalents silica over a seven-day period in an aqueous solution at room temperature. Solid-state silver compounds were identified as silver oxynitrate (Ag$_7$NO$_{11}$, inverted red triangles), silver sulfate (Ag$_2$SO$_4$, green circles), and argentic oxide (AgO, blue squares).

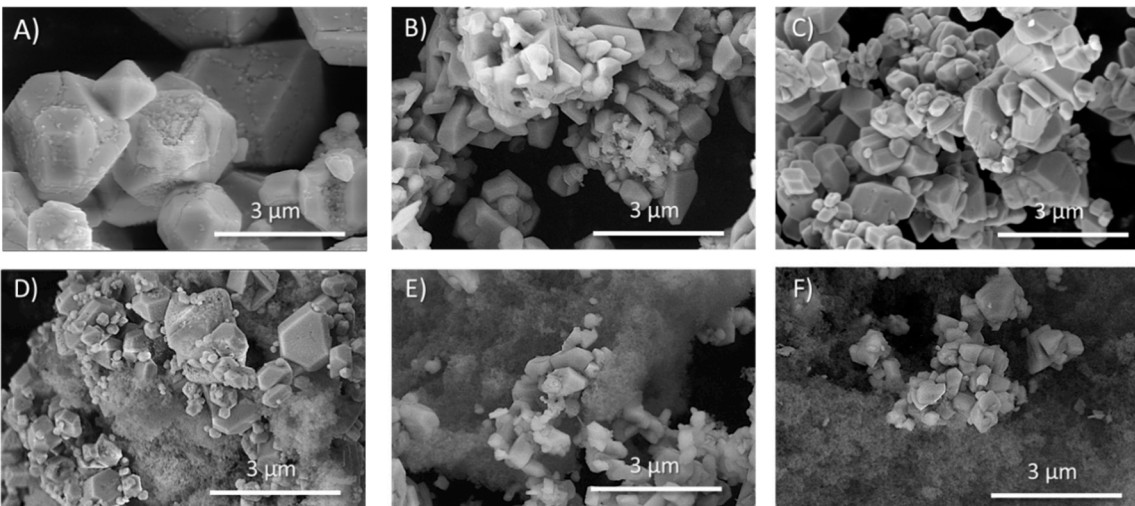

**Figure 7.** Scanning electron microscopy images of silver oxynitrate and Ag$_7$NO$_{11}$:SiO$_2$ containing 0.1:1, SiO$_2$:Ag, exposed to aqueous media for a period of seven days. Silver oxynitrate (Ag$_7$NO$_{11}$) (**A**) prior to aqueous exposure; (**B**) after 24 h of exposure to aqueous media; and (**C**) after 168 h of exposure to aqueous media. Ag$_7$NO$_{11}$:SiO$_2$ containing 0.1:1, SiO$_2$:Ag (**D**) prior to aqueous exposure; (**E**) after 24 h of exposure to aqueous media; and (**F**) after 168 h of exposure to aqueous media.

### 3.4. Antimicrobial Efficacy Evaluation of Ag$_7$NO$_{11}$:SiO$_2$

Silver oxynitrate is known as a potent antimicrobial agent against planktonic, biofilm, and drug resistant bacterium [16–18]. Modification of the form or function of silver oxynitrate may result in an impairment of this antimicrobial activity. Therefore, it is imperative to determine the impact of synthetic modifications on the efficacy of silver oxynitrate. Towards this, a series of antimicrobial methodologies were employed to evaluate the efficacy of silver oxynitrate and Ag$_7$NO$_{11}$:SiO$_2$ against both planktonic and biofilm states of *Staphylococcus aureus* (ATCC 6538) and *Pseudomonas aeruginosa* (ATCC 9027).

Single time point log reduction of Gram-positive *S. aureus* (4 h) and Gram-negative *P. aeruginosa* (1 h) were evaluated in Mueller Hinton Broth (MHB) inoculated to a concentration of $1 \times 10^6$ CFU/mL. The efficacy of silver oxynitrate and $Ag_7NO_{11}:SiO_2$ containing 0.1:1 $SiO_2$:Ag were evaluated at normalized silver concentration. Following treatment, the remaining bacteria were numerated and log reduction quantified versus untreated control. The efficacy of $Ag_7NO_{11}:SiO_2$ was compared with silver oxynitrate and was found to be equivalent for both organisms tested (*p* value > 0.05), as shown in Figure 8A.

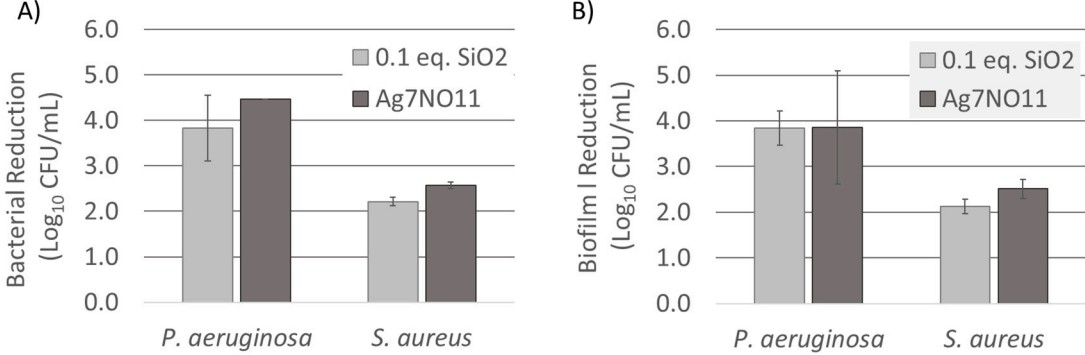

**Figure 8.** Antimicrobial efficacy of silver oxynitrate ($Ag_7NO_{11}$) and $Ag_7NO_{11}:SiO_2$ containing 0.1:1, $SiO_2$:Ag. (**A**) Single-time log reduction for 1-h treatment period against *P. aeruginosa* and 4-h treatment period against *S. aureus* challenged at $1 \times 10^6$ CFU/mL; (**B**) Biofilm log reduction values for 2-h treatment period against *P. aeruginosa* and 4-h treatment period against *S. aureus* established 3D biofilm. Results representing the average of triplicate data (*n* = 3), error bars indicated represent standard deviations of the triplicate measurements.

The role of biofilm in infection and delayed wound healing is becoming increasingly evident [33–35]. Using a clinically relevant biofilm model, adapted from literature, we evaluated the capacity of silver oxynitrate and $Ag_7NO_{11}:SiO_2$ containing 0.1:1 $SiO_2$:Ag to target and reduce biofilm [17,36]. In brief, established biofilms of *P. aeruginosa* and *S. aureus* were treated with silver oxynitrate or $Ag_7NO_{11}:SiO_2$. Following a 2-h and 4-h treatment period for *P. aeruginosa* and *S. aureus* respectively, viable cells were recovered and numerated versus untreated control to determine log reduction values. Silver oxynitrate and $Ag_7NO_{11}:SiO_2$ were found to equivalently reduce viable biofilm counts (*p* value > 0.05), as shown in Figure 8B. In comparison with silver oxynitrate, no hindrance of antibacterial or antibiofilm activity was observed with the incorporation of silver oxynitrate to the silicon dioxide framework through the described co-deposition procedures.

## 4. Discussion

Amorphous silicon dioxides are known to confer desirable attributes upon incorporation with active ingredients including enhanced biocompatibility, improved chemical stability, and improved thermal stability [7,9,37–39]. Modified Stöber methodologies employed by Cong et al. afford silica coatings on fluorescent dye-doped polymeric nanoparticles towards obtaining an insulating layer with improved physical stability [37]. Silver-silica core-shell structures, successfully prepared by Xu et al., use a modified Stöber process to confer chemical stability to metallic silver nanoparticles without hindering antimicrobial efficacy [9]. These methods are conducive to conventional organic drugs or weak oxidizing agents such as metallic silver, affording control over polymerization and deposition of silicon dioxide. However, in the presence of strong oxidizing agents such as silver oxynitrate, chemical incompatibilities restrict the successful application of these methodologies. Acid-catalyzed polymerization or deposition of alkali silicates are also known to generate amorphous silica [5,24,25,27]. Similar to the aforementioned silica-shell strategies, conventional acid-precipitated silicate methodologies are reliant upon the use of surfactants and organic precursors as structure-directing agents and therefore pose challenges for use with highly oxidizing materials. In the present work, alternate methods for the in-situ polymerization

of silica with higher oxidation state silvers are presented wherein the formation of the silica structure is self-directed through acidic by-products of silver oxynitrate synthetic reactions.

Silicate ions will polymerize to form silica in solutions having pH of less than about 10 ($H_4SiO_4$, $pK_{a1}$ = 9.8) [40]. In the preparation of silver oxynitrate, Equation (3), it is known that sulfuric and nitric acids are generated as by-products of the silver oxidation reaction resulting in the reduction of pH of the solution from neutral to pH 1–2 (data not shown). These acids, as shown previously, are suitable to effect protonation of silicic acid at room temperature and formation of oligomeric silicic acids resulting in the precipitation of amorphous silica [24,25,40]. In the methodologies presented herein alkali silicates at various concentrations relative to silver are dispersed into an oxidizing solution of potassium persulfate to which a soluble silver solution is added resulting in the oxidation of silver and chemical deposition of silver oxynitrate, as verified by XRD in Figure 1. Finite nucleation locus of silver oxynitrate, a crystalline compound with the cubic space group Fm3m, generate an acid gradient in the immediate vicinity of silver oxynitrate nucleation sites, directing the spatial polymerization or precipitation of silica [41]. As observed in Figures 2 and 3, silver oxynitrate solids are observed to be impregnated in a silica framework. Absence of AgO or other degradation products of silver oxynitrate, as seen in in Figure 1, indicates that the co-deposition process does not impair the formation of silver oxynitrate. Formation of silver sulfate does not occur as a by-product of the reaction, as seen in Equation (3). Rather silver sulfate is an impurity forming due to incomplete conversion of soluble silver to silver oxynitrate. This impurity is a minor component in the silver oxynitrate control and observed at increasing relative quantities as silica ratios rise from 0.1:1 to 0.5:1, $SiO_2$:Ag, as shown by relative peak area in Figure S1.

Increasing the relative concentration of silica during the co-deposition reaction was also shown to be an effective method for crystalline size control of silver oxynitrate. Utilizing the FWHM for the four strongest reflections in the diffraction pattern for silver oxynitrate, observed at 31.2, 36.3, 52.3, 62,2 °2Θ, respectively (222), (400), (220) and (622) reflections, the relative size of the silver oxynitrate crystals were calculated as per the Debye-Scherrer equation [41,42]. Based upon the average crystal size determination from these primary reflections, the crystal size of silver oxynitrate decreases from 1032 ± 152 Å to 500 ± 75 Å as the relative ratio of $SiO_2$:Ag increases from 0.0:1 to 0.5:1. This same trend in silver oxynitrate crystalline size reduction upon increasing silica ratio may be observed the SEM and TEM images collected for the co-deposition products as shown in Figures 2 and 3.

As silica is known to confer enhanced stability of active ingredients, the stability of silver oxynitrate and $Ag_7NO_{11}$:$SiO_2$ were compared over the course of four months under ambient and 40 °C storage. The thermal degradation pathways for silver oxynitrate are well known [21,23,30]. The first two decomposition processes for silver oxynitrate, Equations (4) and (5) are exothermic with associated enthalpies of ΔH = −244 KJ/mol and −75.8 KJ/mol and initiation temperatures of 80–85 °C and 115–120 °C respectively [21]. In contrast, the subsequent degradation processes, Equations (6) and (7), are endothermic with associated enthalpies of ΔH = 659 KJ/mol and 338 KJ/mol and initiation temperatures of 350 °C and 425–435 °C respectively [21]. It is also known that the rate of decomposition of silver oxynitrate is accelerated in the presence of heat and water [30,41]. Under ambient conditions, silver oxynitrate is observed to proceed through the first decomposition process as shown in Figure 4A. The rate of this degradation process is retarded in $Ag_7NO_{11}$:$SiO_2$ as shown in Figure 4B. At elevated temperatures, a more rapid degradation of silver oxynitrate will result in increased thermal energy due to the exothermic nature of the first two degradation processes. In combination with the elevated storage temperature, initiation of subsequent endothermic degradation processes as shown in Equations (6) and (7) are observed to occur for silver oxynitrate as seen in Figure 5A. These advanced degradation processes are not observed in $Ag_7NO_{11}$:$SiO_2$. At elevated temperatures only the first degradation process, Equation (4), is observed for $Ag_7NO_{11}$:$SiO_2$ as seen in Figure 5B.

Silica is a thermal insulator with a very low thermal conductivity; beneficial in aerospace and construction applications [43–45]. As an aerogel, Reim et al. demonstrated silica granulates may afford a heat transfer coefficient of less than 0.4 W/($m^2$ K) [44]. Silica is also well known as a desiccant, capable

of sequestering water from organic solvents to less than 100 ppm and significant reductions in water content in solid-state powder samples [46,47]. The results presented here suggest that the presence of an interspersed silica framework with thermally insulating and desiccating properties may obstruct heat transfer and restrict the step-wise degradation of silver oxynitrate. These properties enhance the long term and thermal stability of silver oxynitrate within the silica framework.

Silver oxynitrate has been previously shown to rapidly and effectively eliminate bacterial organisms in both free and biofilm states [16–18]. Silicate drug delivery systems, such as silica core-shell systems, may provide an enhanced stability and biocompatibility however, they are also known to modify drug release profiles [3,4]. Gaining understanding for how the silica co-deposition process may impact solution-phase behavior, a series of investigations were performed to evaluate silver oxynitrate and $Ag_7NO_{11}:SiO_2$ antimicrobial activities and degradation profiles in aqueous media. As described above, water is known to accelerate the rate of degradation of silver oxynitrate to silver oxide [30,41]. This effect is observed in Figure 6A where over the course of seven days silver oxynitrate proceeds through the degradation process described in Equation (4) to form argentic oxide: AgO. A similar degradation process is observed for $Ag_7NO_{11}:SiO_2$ as shown in Figure 6B. This solid-state transformation is visualized by the change in crystalline morphology by SEM where the cuboctahedra structure of silver oxynitrate observed prior to aqueous exposure, Figure 7A,D is replaced by geometric platelets of monoclinic silver (I, III) oxide over the seven day period [22,41]. Similarly, equivalent antimicrobial and antibiofilm log reduction values ($p$ value > 0.05) are observed for silver oxynitrate and $Ag_7NO_{11}:SiO_2$ against *P. aeruginosa* and *S. aureus* as shown in Figure 8. These results are in agreement with the silica co-deposition framework providing enhanced thermal stability without hindering aqueous degradation profiles or antimicrobial efficacy.

## 5. Conclusions

Methods described herein for the co-deposition of silver oxynitrate-silica in a one-pot synthesis yield a high purity silver oxynitrate within a three-dimensional silica framework. Spatially guided polymerization or deposition of silica, without the aid of alkyl silicates or surfactants, was achieved through the production of acidic by-product generated gradients from silver oxynitrate nucleation sites. Co-deposition of silica, from 0.1:1 to 0.5:1 molar equivalents $SiO_2:Ag$, were shown to influence control over the crystalline size of silver oxynitrate while providing enhanced long term and thermal stability of silver oxynitrate within the silica framework. It was shown that the enhanced thermal stability of silver oxynitrate within the silica co-deposition framework did not hinder aqueous degradation profiles or antimicrobial and antibiofilm activity of silver oxynitrate. The methods described herein will confer increased stability and versatility of silver oxynitrate when incorporated into inherently reducing materials such as natural fibers or hydrogels and during thermal processing as required for materials such as silicones, melt adhesives, or thermoplastics. Providing an opportunity to expand the application of silver oxynitrate into novel functional materials to address the growing need for anti-infective technologies.

**Supplementary Materials:** The following are available online at http://www.mdpi.com/2076-3417/9/23/5236/s1.

**Author Contributions:** C.J.S.: Conceptualization, Methodology, Formal Analysis, Writing—Original Draft, Visualization, Project Administration, Supervision, Funding Acquisition. R.L.: Methodology, Validation, Investigation. C.G.: Methodology, Formal Analysis, Investigation, Data Curation, Writing—Review & Editing. M.H.: Methodology, Validation, Investigation, Formal Analysis. R.P.: Writing—Review & Editing, Supervision, Project Administration, Funding Acquisition.

**Funding:** This research was supported by the BioTalent Canada, Student Work Placement Program (SWPP) between Carlie Gooodall, student of the University of Guelph, and Exciton Technologies Inc.

**Acknowledgments:** The authors gratefully acknowledge the support of the Medical Imaging Laboratories and the Centre for Nanostructured Imaging at the University of Toronto in Toronto, Ontario.

**Conflicts of Interest:** Carla J. Spina, Michelle Hay, Roohee Ladhani, Carlie Goodall, and Rod Precht were employees of Exciton Technologies Inc. at the time the work was completed.

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
