# Peer review of "Directed Silica Co-Deposition by Highly Oxidized Silver: Enhanced Stability and Versatility of Silver Oxynitrate"

_applsci, doi:10.3390/app9235236_

Round 1

Reviewer 1 Report

The manuscript is interesting and well written.

There are some minor corrections, as listed below. As a general comment, the text could be shortened, the experimental procedure is very well explained, with repetitions in all captions and in the text. In my view this is superfluous and some technical detail, well described in the experimental, could be removed from the main text and captions.

1) introduction is very rich of details, some revision to made it a bit more concise is suggested. The same holds for the discussion; the first two/three paragraphs (ultil line 23 of page 25) are again a review of literature, with different details with respect to the introduction but anyway maybe a bit too extended for a regular paper. I suggest some revision to make it more concise also in this part

2) There is some inconsistency in the caption of Figure 3 and discussion in the text. Please check

3) Line 24 of page 7: ' the silver oxynitrate decreases from 0:0 to 0.5:1 SiO2:Ag'. Please check: a ratio of 0:0 does not make any sense.

4) The way the samples are referred to in the text (e.g. 0.1:1 SiO2:Ag silver oxynitrate co-deposition products' could be made shorter, to simplify the text and make it more readable. Also, referring to the samples as 'products' is confusing, since in some parts of the text decomposition products are discussed.

5) The XRD discussion is very detailed and could be shortened.

Reviewer 2 Report

The manuscript reported a one-pot method to deposition silver oxynitrate as the core in silica shell framework. The silica framework was showed to reduce thermal degradation but not affect the antimicrobial activity of silver oxynitrate. The experimental design and writing are good. There are some comments as below.

Please check the format throught the manuscript. E.g. Line 23, a punctuateon is lacking before “Addressing”. Figure 2 caption, “(e-r)” did not marked in Figure 2, please correct it. How was the morphology changing during thermal decomposition of silver oxynitrate and silver oxynitrate-silicon dioxide co-deposition products under elevated temperatures? Please provide SEM characterization and discuss it.
